# Social Determinants of Health and College Food Insecurity

**DOI:** 10.3390/nu16091391

**Published:** 2024-05-04

**Authors:** Catherine Mobley, Ye Luo, Mariela Fernandez, Leslie Hossfeld

**Affiliations:** 1Department of Sociology, Anthropology and Criminal Justice, Clemson University, Clemson, SC 29634, USA; yel@clemson.edu; 2Department of Parks, Recreation and Tourism Management, Clemson University, Clemson, SC 29634, USA; mfrnndz2@illinois.edu; 3College of Behavioral, Social and Health Science, Clemson University, Clemson, SC 29634, USA; lhossfe@clemson.edu

**Keywords:** childhood food insecurity, college student food insecurity, health and well-being, social determinants of health (SDOH), stress

## Abstract

In recent years, many students have faced economic hardship and experienced food insecurity, even as universities strive to create more equitable pathways to college. There is a need for a more holistic perspective that addresses the complexity of food insecurity amongst college students. To this end, we examined the relationship between the social determinants of health, including college food insecurity (CoFI) and childhood food insecurity (ChFI), and their relationship with well-being measures. The study sample was a convenience sample that included 372 students at a public university who responded to an online survey in fall 2021. Students were asked to report their food security status in the previous 30 days. We used the following analytical strategies: chi-square tests to determine differences between food secure (FS) and food insecure (FI) students; binary logistic regression of CoFI on student demographics and ChFI; and ordinal or binary logistic regression for well-being measures. Black students, off-campus students, first-generation students, in-state students, and humanities/behavioral/social/health sciences majors were more likely to report CoFI. FI students were more likely to have experienced ChFI and to have lower scores on all well-being measures. ChFI was associated with four well-being measures and its effects were mediated by CoFI. College student health initiatives would benefit from accounting for SDOH, including ChFI experiences and its subsequent cumulative disadvantages experienced during college.

## 1. Introduction

Food security refers to having food “access by all people at all times, to enough food for an active, healthy life” [1] (p. 2); such access should be “consistent” and “dependable” [1] (p. i). Food insecurity pertains to the “limited or uncertain availability of nutritionally adequate and safe foods or limited or uncertain ability to acquire acceptable foods in socially acceptable ways” [2] (p. 3) and the inability to afford and have access to healthy food, such as fresh produce [3].

Many students face economic hardship and experience food insecurity, even as universities strive to create equitable pathways to college. There has been increased concern about the ability of college students to afford school [4,5]. A national report that assessed basic needs insecurity across community colleges and four-year institutions in the U.S.A. found that 48% of students attending two-year institutions and 41% of students attending four-year universities were food insecure [6]. The rates of food insecurity were also documented in a national representative survey of approximately 100,000 college students conducted by the National Center for Education Statistics in Spring 2020. According to McKibben et al.’s analysis of this data, 23% of undergraduate students and 12% of graduate students said they experienced low or very low food insecurity. These rates are higher than the 2020 national food insecurity rate of 10.5% [7].

Research on food insecurity often aggregates across all adults living in typical households. However, there is a need for a more holistic perspective that addresses the complexity of food insecurity amongst college students. Food insecurity may take on different dimensions in the lived experiences of college students that may not be adequately captured by USDA measures of food insecurity that focus on a household context. This study presents a case for utilizing the Social Determinants of Health (SDOH) framework to assess and address college food insecurity. College represents a distinct transition for students into independence (or continued dependence) and a continuation of advantages (or disadvantages) from childhood. However, there is limited research that empirically examines the relationship between childhood food insecurity (ChFI) and college food insecurity (CoFI) among college students and their joint impact on various measures of college student health and well-being. Our study aimed to fill these gaps by using an SDOH framework to identify which sociodemographic groups of college students were more likely to experience college food insecurity and to assess whether and how college food insecurity is associated with prior experiences with food insecurity. We also investigated how childhood and college food insecurity were associated with several health and well-being measures. This research helps to advance understanding of the relationship between food access and college students’ well-being and provides valuable information for the development of a more holistic approach to college student health and well-being and for the design of effective interventions and policies that can help reduce college food insecurity.

### 1.1. Social Determinants of Health (SDOH) Framework

Much research on food insecurity among college students focuses on individual-level explanations and solutions, i.e., what students themselves can do to avoid food insecurity or how they may have contributed to their own food insecurity. For example, university officials have recently identified students’ lack of awareness about service availability, feelings of stigma or shame, low levels of financial literacy, an unwillingness to accept help, and difficulty in complying with program eligibility requirements as the five primary challenges related to meeting students’ basic needs [8]. However, the causes of college food insecurity run deeper and are related to structural and institutional factors. Any solution must address this complexity, not just by ensuring access to food, but by addressing the root causes of food insecurity.

The Social Determinants of Health (SDOH) approach provides a valuable theoretical framework that captures the complex roots of food insecurity and enables a more equitable perspective to address this challenge on college campuses. The SDOH approach addresses “the conditions in which people are born, grow, live, work and age” [9], and how these conditions influence health outcomes. The U.S. Department of Health and Human Services [10] identified five dimensions of SDOH: (1) economic stability; (2) education access and quality; (3) health care access and quality; (4) neighborhood and built environment; and (5) social and community context. A robust SDOH framework incorporates the concept of structural competency, the recognition that systemic inequalities serve to perpetuate health disparities across time and place. Structural competency “shifts the gaze away from individual, ‘cultural’ responsibility, like food choice, towards the ways in which health outcomes are impacted by the organization of institutions and policies, like food access” [11] (p. 10). A central tenet of the SDOH framework is that health determinants vary within and across social groups and accumulate and intensify (or are reduced and dissipate) throughout one’s life [12]. Many college students experience transitional instability that makes them particularly vulnerable to experiencing unmet SDOH needs [13].

There is a growing body of research using the SDOH framework to study college student experiences [13,14,15,16,17]. SDOH is a valid framework for assessing college students’ health and well-being [18] and expanding understanding of food insecurity in general [19] and of college student food insecurity, including its relationship with mental health [20]. SDOH for college students includes food, housing, and financial stability [13]. College students’ food choices are shaped by structural inequality. Students may desire to have access to food on a consistent basis and to make healthier choices; however, such access and choices are often not readily available. For example, Agyemang’s [21] study highlighted the structural barriers that led to food insecurity of racialized international students, who were unable to work and subsequently to access culturally appropriate foods. As evidenced by SDOH, research suggests that universities are responding to this call by developing more comprehensive campus health initiatives that more adequately address health disparities [22]. Below, we examine two dimensions of SDOH for college students, including the sociodemographic determinants of college food insecurity and the influence of childhood food insecurity on college food insecurity.

### 1.2. Sociodemographic Characteristics and College Food Insecurity

It is essential to consider the intersectional nature of inequality and social determinants of health, both temporally and spatially. For example, students from groups that have been historically disadvantaged, including racial and ethnic minorities and first-generation students, have higher risks of college food insecurity [23,24,25]. A meta-analysis of 47 studies on college student food insecurity revealed that female students, Black students, first-generation students, and living off campus was associated with higher rates of food insecurity [26]. Transgender and non-binary students are more likely than self-identified male- and female-gendered students to experience food insecurity [27]; LGBTQIA students are at higher risk of food insecurity [28]. As Camelo and Elliott [29] contended, such outcomes for marginalized groups reflect “cumulative disadvantages these students have faced from lifetime exposure to food insecurity and unequal access to the opportunity to learn in the public K–12 school system” (p. 309).

### 1.3. The Influence of Childhood Food Insecurity on Adult Food Insecurity

The stigma-reducing approach of SDOH focuses on social justice to increase well-being and retention among college students and incorporates a rights-based focus on the social determinants of health more broadly [30] to also include prior experiences with food insecurity. The cumulative disadvantage/advantage hypothesis refers to “the systemic tendency for interindividual divergence in a given characteristic (e.g., money, health, or status) with the passage of time” [31] (p. s327). In the context of our study, this framework highlights that students are not blank slates when they arrive at college. They are shaped by their prior experiences, including food insecurity, depression, emotional and physical abuse, housing insecurity, and racial and ethnic discrimination [32]. These prior experiences then shape a person’s ability to navigate life transitions [33], such as moving from one’s childhood situation to attending college. It is also important to note that at the meso-level, institutions themselves may reinforce cumulative disadvantages [34]; in the context of this study, this would include colleges and universities and the ways in which they do, or do not, adequately address student health and well-being.

Individuals who experience economic disadvantages during childhood are likely to experience further disadvantages after becoming adults. Early life stressors that occur during key developmental stages can have long-lasting impacts on mental health and other outcomes [35]. Although there is strong evidence that childhood food insecurity is associated with child health outcomes [36,37], there is little research that linked childhood food insecurity and college students’ health and well-being. The limited prospective longitudinal research shows that a lack of sufficient food during childhood is associated with greater psychological distress [38] and poorer general health [39] in young adulthood. Previous research also showed a relationship between childhood experiences of food insecurity and adult food insecurity experiences [40], including college students’ experiences [41,42]. Students from households that experience food insecurity are more likely to experience food insecurity in college [43]. Experiences with food insecurity in one’s family of origin may shape experiences with, or perceptions of, food insecurity as an adult [44,45]. This impact is related to one’s experience with early childhood financial adversity, in that such adversity is often associated with a disadvantaged financial status in adulthood, which may lead to fewer resources for healthy eating. Individuals who experienced childhood food insecurity may also develop less healthy eating behaviors as an adult [45,46]. Nettle et al.’s [47] study showed that individuals who experienced food insecurity as a child may behave as if they are food insecure as an adult, even if they are not actually food insecure, pointing to the long-term psychological impacts of childhood food insecurity reported in other studies [44,45].

### 1.4. Food Insecurity, Health and Well-Being, and Academic Progress of College Students

Attending college is a major transition for many students, whether they are a traditional college student, between the ages of 18–22 who is living on their own and taking care of their own needs for the first time, or a non-traditional college student who may be balancing the competing demands of family, work, and school. These challenges further illustrate the relationship between SDOH and negative outcomes for college students, including outcomes related to hunger and food insecurity [48,49]. An essential task for these students is ensuring they can have access to healthy food on a regular basis. Many students must learn to make independent decisions about food choices, meal preparation, and budgeting for a healthy diet. Students facing food insecurity may experience peer pressure and heightened awareness of the stigma associated with food insecurity [50], making it less likely they will talk to others about their experiences, and even less likely they will actively seek help, such as applying for SNAP or visiting a food pantry, especially a campus-based food pantry [51]. Furthermore, students are figuring out how to ensure they are successful in school while also attending to their physical and mental health.

College is a particularly stressful time for many students, as stress impacts academic performance and can lead to changes in food consumption patterns [52]. Experiencing food insecurity in college results in more negative physical health and mental health outcomes [53,54] and poorer academic outcomes [53,55,56] for students. Students often struggle to maintain a healthy diet as they face a variety of stressors [57]. College students tend to have less healthy diets, relying on convenience meals, fast food, and less expensive options [52]. These practices often result in food insecurity, which can, in turn, cause stress and anxiety [58]. Food-insecure students indicate that a major source of anxiety is their inability to afford healthy food [3], which results in increased stress [59], difficulty concentrating on one’s studies [60], and lower grades [23]. Food insecurity ultimately impacts degree attainment, with food-insecure students having lower odds of graduating [61]. This is especially true for first-generation college students who experience food insecurity [62].

Students’ challenges with food insecurity were compounded during the COVID-19 pandemic [63]. One study found that slightly more than one-third of college students at one university were experiencing food insecurity within a 30-day period during the pandemic [16]. Just over 15.1% of students were newly food insecure after the onset of the pandemic [64]. Soldavini et al. [65] found that while approximately two-thirds of students reported their food access remained unchanged during the pandemic, 28% reported that access became worse after the onset of the pandemic. Another study revealed that during the pandemic, nearly one-fourth of students reported that the quality of their food had gotten worse during the pandemic and 12% indicated that they regularly missed meals because they could not always afford food [66].

### 1.5. Research Hypotheses

Based on the SDOH framework and the literature reviewed above, we proposed the following hypotheses:

**Hypothesis** **1.**
*College students’ sociodemographic characteristics and childhood food insecurity experience affect their college food insecurity experience. More specifically, female students, Black students, first-generation students, students living off campus, students from the LGBTQIA++ community, and students with childhood food insecurity experience are more likely to experience college food insecurity.*


**Hypothesis** **2.**
*College food insecurity is negatively associated with perceived health and well-being outcomes; as compared with college students who do not experience food insecurity in college, college students who experience food insecurity will perceive they have poorer health and well-being than students who do not experience food insecurity in college.*


**Hypothesis** **3.**
*Retrospectively ascertained childhood food insecurity affects college students’ health and well-being, and this effect is mainly mediated by college food insecurity.*


## 2. Materials and Method

### 2.1. Procedure and Participants

Our study took place at a public, land grant university in the Southeastern United States. In 2021, the student enrollment was approximately 27,000 (79.2% undergraduates, 20.8% graduate students). The university has one student-run, student-led food pantry. This study received Institutional Review Board approval in October 2021. Data collection took place during October and November 2021. Prior research supports the implementation of surveys on food insecurity of college students during the fall semester vs. the spring semester, as students have lower fall-to-spring persistence rates [67]; thus, surveys in the fall tend to capture more students. To participate, students had to be currently enrolled during the fall 2021 semester.

The survey was publicized through classroom visits from the research team, emails to faculty, invitations sent to friends and colleagues, and flyers and newsletters. We provided individuals with a link to an anonymous Qualtrics survey. As an incentive, students could enter their name into a draw for one of 90 gift certificates (i.e., fifty USD 10 gift cards, thirty USD 25 gift cards, ten USD 50 Amazon gift cards). At the end of the survey, respondents exited the primary survey and were directed to a new survey, where they could enter their name into the draw.

### 2.2. Measures

The survey was designed to capture data about several dimensions of food access and food insecurity for university students. We included sociodemographic variables, measures of college and childhood food insecurity, and six measures of health and well-being.

#### 2.2.1. College Food Insecurity

We used the following six items from the USDA’s [68] U.S. Household Food Security Survey to assess the students’ food security status: In the last 30 days: (1) The food that I bought just didn’t last and I didn’t have money to get more. (2) I couldn’t afford to eat balanced meals. (3) Did you ever cut the size of your meals or skip meals because there wasn’t enough money for food? (4) How often did this happen? (5) Did you ever eat less than you felt you should because there wasn’t enough money for food? (6) Were you ever hungry but didn’t eat because you didn’t have enough money for food? Answering “sometimes true” or “often true” to items 1 and 2; “yes” to items 3, 5, and 6; and “almost every day” or “2–3 days” to item 4 were considered as affirmative responses for these items.

Students providing affirmative responses to two or more of these items were classified as “food insecure”; all others were classified as “food secure”. As we were most interested in assessing the experiences of college students while school was in session, we asked them to report on their experiences in the previous 30-day period, which was an approach taken in prior studies on college student food insecurity [40]. This strategy reduces the possibility of response bias associated with longer timeframes and was used in studies of college food insecurity during the COVID-19 pandemic [16,63].

#### 2.2.2. Childhood Food Insecurity

Based on prior research [69], we included the Hunger Vital Scale, which comprises two survey items to ascertain childhood food insecurity, asking whether the following statements were true during their childhood: (i) “In my family, the food that we bought just didn’t last, and we didn’t have money to get more” and (ii) “In my family, we worried whether our food would run out before we got more money to buy more”. Because only a small number of students answered “often true” or “sometimes true” to each question (34 for food not lasting and 37 for food running out), we created a dichotomous measure of childhood food insecurity. A student was identified as experiencing childhood food insecurity if they answered “sometimes true” or “often true” to one or both questions. The Hunger Vital Sign has been used in diverse contexts, including to assess food insecurity among college students [70,71]. Our study used this measure in a retrospective fashion (i.e., asking students to recall whether the two items reflect their childhood experiences). Thus, our use of this measure was exploratory. Despite these limitations, our study established a foundation for future investigations using retrospective measures of food insecurity. Such studies should also focus on concurrent (and potentially, reversible) conditions that impact food insecurity.

#### 2.2.3. Food Stress

Students were asked “How would you rate the level of stress you have experienced, related to obtaining enough food to eat, during the current semester”? The 5-point response scale ranged from “no stress at all” to “tremendous stress”. Because very few students answered “tremendous stress”, we combined this category with “more than average stress”. This question was modified from a prior study on food insecurity among college students [71].

#### 2.2.4. Difficulty Concentrating on Studies Due to Food

Students who answered “yes” to the question “At any time this semester, did you have difficulty concentrating on your studies because you did not have enough money for food”? were compared with those who answered “no”. This question was modified from a prior study on food insecurity among college students [71].

#### 2.2.5. General Stress

Students were asked “How would you rate the overall level of stress you have experienced during the current semester”? The 5-point response scale ranged from “no stress at all” to “tremendous stress”. Because very few students answered “no stress”, we combined this category with “less than average stress”. This question was modified from a prior study on food insecurity among college students that used a timeframe of the last 12 months [71].

#### 2.2.6. Self-Rated Health

Students were asked “How would you describe your general health”? with the 5-point response scale ranging from “poor” to “excellent” [71]. Because very few students answered “poor”, we combined this category and “fair”.

#### 2.2.7. COVID’s Impact on Access to Food

We asked students “How would you compare the challenges that you faced obtaining food prior to the onset of COVID-19 (i.e., approximately March 2020) with the challenges that you faced after the onset of COVID-19”? We compared students who answered that they had faced more challenges after the onset of COVID-19 with those who answered that they had faced fewer or similar challenges after the onset of COVID-19. This question was modified from our prior studies on food insecurity in rural South Carolina after the onset of the global pandemic [72].

#### 2.2.8. Perceived Academic Progress

Students rated their overall progress in school, including graduating on time; the 4-point response scale ranged from “poor” to “excellent” [73].

#### 2.2.9. Demographic Characteristics

We included several students’ self-reported demographic characteristics: gender (female = 1, male or other = 0), sexual orientation (transgender, gay, lesbian, bisexual, asexual, questioning, or other term = 1; heterosexual = 0), race/ethnicity (non-Hispanic White, non-Hispanic Black, non-Hispanic other race/ethnicity, multiracial, or Hispanic; four dummy variables with non-Hispanic White as the reference category), living situation (off campus = 1 or on campus = 0), and whether they were a first-generation college student (yes = 1 or no = 0). We also included where they came from (in state = 1 or out of state = 0) and their field of study (agriculture/engineering/science, humanities/behavioral/social/health/education, and business; two dummy variables with agriculture/engineering/science as the reference category) as additional demographic controls. For the question on gender, 3 students chose “non-binary” and 1 chose “prefer not to answer”. They were coded 0 for the female dummy variable and were coded 1 for the sexual orientation variable. Additional analysis with these 4 cases removed produced similar results.

#### 2.2.10. Analytical Strategy

A total of 449 students completed the survey. We excluded 34 students who were missing data on college food security questions. An additional 36 students were excluded due to missing data on gender and 7 more students were excluded due to missing data for other variables. The analytical sample included 372 students.

To answer our research questions, we first calculated descriptive statistics for the whole sample and then separately for students who were identified as food secure and students who were identified as food insecure in college. We tested whether there were statistically significant differences between these two groups using a chi-square test (two-tailed). We then ran a binary logistic regression of college food insecurity on student demographics and childhood food insecurity. We estimated two models. Model 1 included only student demographics; model 2 added childhood food insecurity, allowing us to determine whether childhood food insecurity explained the associations between student demographics and college food insecurity.

Next, we ran ordinal or binary logistic regression to examine the effects of childhood and college food insecurity on the health and well-being measures. For each health and well-being outcome, we estimated two models. Model 1 included student demographics and childhood food insecurity, and model 2 added college food insecurity. We tested whether childhood food insecurity indirectly affected each outcome through college food insecurity with the KHB method developed by Karlson, Holm, and Breen [74]. The method decomposes the difference in the coefficients of an independent variable between models with and without the mediator into the part attributable to the mediator (indirect effect) and the part attributable to the rescaling of the coefficient that occurs across nested nonlinear probability models. In addition, because our study tested the effects of food insecurity on multiple health and well-being outcomes simultaneously, our *p*-values from ordinal regressions may be underestimated. As a robust check, in addition to the *p*-values calculated within each individual regression, we also calculated the Romano–Wolf stepdown-adjusted *p*-values, which were corrected for multiple hypothesis testing [75]. We conducted analyses with STATA (18.0, StataCorp LLC, College Station, TX, USA). We considered an effect statistically significant if *p* < 0.05 in the two-tailed tests. Because our hypotheses were directional, we also note the effects that were marginally significant at *p* < 0.1 in the two-tailed tests.

## 3. Results

### 3.1. Sample Description and Results of Chi-Square Tests

Table 1 reports descriptive statistics for the entire sample and for food-secure and food-insecure students separately. Overall, the majority of respondents were females (73%); about 12% identified themselves as LGBTQIA++. The majority identified as non-Hispanic White (77%). Approximately 8% identified themselves as multiracial and another 8% as Hispanic. Non-Hispanic Black students accounted for 3% and other races/ethnicities combined accounted for less than 5%. Overall, 23.7% of the students reported they faced food insecurity.

When comparing food-insecure students with food-secure students, the proportions of students who were living off campus (71% vs. 51%), in-state students (76% vs. 56%), first-generation college students (21% vs. 8%), and majoring in humanities/social/behavioral/health sciences (60% vs. 43%) were higher among those who were food insecure than those who were food secure; these differences were statistically significant (*p* < 0.01 for all, except for study area, where *p* < 0.05). Also, higher among food-insecure students were the proportions of students who had childhood food insecurity experience (27% vs. 7%), who had difficulty concentrating on their studies due to food problems (22% vs. 2%), and who stated they faced more food challenges after the outbreak of the COVID-19 pandemic (32% vs. 14%). Their mean levels of food stress (2.78 vs. 1.57) and general stress (3 vs. 2.58) were higher, while their self-rated health (2.17 vs. 2.60) and perceived academic progress were lower (3.23 vs. 3.49) than food-secure students. All differences were statistically significant (*p* < 0.01 for all, except for perceived academic progress, where *p* < 0.05).

### 3.2. Student Demographic Characteristics, Childhood Experience of Food Insecurity, and College Food Insecurity

Binary logistic regression of college food insecurity on student characteristics and childhood food insecurity was used to test our first hypothesis and the results are presented in Table 2. Controlling for other demographic characteristics in the model for college food insecurity, students living off campus (AOR = 1.98, 95% CI = [1.14, 3.44], *p* < 0.05), in-state students (AOR = 1.82, 95% CI = [1.02, 3.26], *p* < 0.05), first-generation students (AOR = 2.29, 95% CI = [1.08, 4.85], *p* < 0.05), and humanities/social/behavioral/health sciences students (AOR = 2.09, 95% CI = [1.16, 3.78], *p* < 0.05) were more likely to report college food insecurity (Table 2, model 1). Non-Hispanic Black students were much more likely than White students to report college food insecurity, though this difference was only marginally significant (AOR = 3.44, 95% CI = [0.90, 13.16], *p* < 0.1). Students who experienced childhood food insecurity were more than three times as likely to report college food insecurity as students who did not experience childhood food insecurity (AOR = 3.17, 95% CI = [1.541, 6.507], *p* < 0.01; Table 2, model 2). The associations of non-Hispanic Black in-state and first-generation college students with food insecurity were substantially attenuated and were no longer statistically significant after childhood food insecurity experience was added in model 2, meaning childhood food insecurity (or associated factors, such as socioeconomic status) explained much of these relationships.

### 3.3. Food Insecurity and Health and Well-Being

To examine how students’ food experiences in childhood and college affected health and well-being, we conducted binary and ordered logistic regressions of the well-being and health measures. For each outcome, we present two models: model 1 with student demographic characteristics and childhood food insecurity experience; model 2 added the college food insecurity measure to model 1. These models tested our third hypothesis of the effect of childhood food insecurity on college students’ health and well-being and the mediating effect of college food insecurity in this relationship. Model 2 tested our second hypothesis on the effect of college food insecurity on health and well-being measures.

As shown in Table 3 and Table 4, when controlling for student demographic characteristics, students who experienced childhood food insecurity were more likely to experience a higher level of food stress (AOR = 2.62, 95% CI = [1.39, 4.92], *p* < 0.01) and face greater challenges accessing food after the COVID-19 outbreak (AOR = 2.00, 95% CI = [0.93, 4.30], marginally significant *p* < 0.1), more likely to have difficulty concentrating on their studies due to food (AOR = 5.25, 95% CI = [1.97, 13.95], *p* < 0.01), more likely to have a higher level of general stress (AOR = 2.03, 95% CI = [1.10, 3.74], *p* < 0.05), and less likely to have higher levels of self-rated physical health (AOR = 0.39, 95% CI = [0.21, 0.70], *p* < 0.01) compared with students who did not have childhood food insecurity experience (model 1). After college food insecurity was added in model 2, the effect of childhood food insecurity became non-significant for food stress, COVID-19’s impact on food access, and general stress, suggesting the effect of childhood food insecurity was largely mediated by college food insecurity experience. However, even after controlling for college food insecurity, childhood food insecurity remained a strong and significant predictor of difficulty concentrating on studies (*p* < 0.01) and self-rated health (*p* < 0.05), suggesting that for these outcomes, childhood food insecurity was associated with these outcomes after their indirect effects through college food insecurity were considered. Mediation analysis using the KHB method showed that for all outcomes, the indirect effects of childhood food insecurity through college food insecurity were statistically significant. The proportion of the total effects that were mediated by college food insecurity ranged from 23% to 56% (see Appendix A).

College food insecurity was strongly associated with all six health and well-being outcomes; when controlling for student demographic characteristics and childhood food insecurity, students who experienced food insecurity in college were more likely to have higher levels of food stress (AOR = 10.87, 95% CI = [6.37, 18.54], *p* < 0.01), more likely to experience difficulty concentrating on their studies (AOR = 11.55, 95% CI = [3.81, 35.05], *p* < 0.01), more likely to face greater challenges accessing food after the COVID-19 outbreak (AOR = 2.54, 95% CI = [1.36, 4.74], *p* < 0.01), and more likely to have higher levels of general stress (AOR = 2.80, 95% CI = [1.67, 4.71], *p* < 0.01) than students who did not have food insecurity experience in college. They were less likely to have higher levels of self-rated health (AOR = 0.42, 95% CI = [0.26, 0.69], *p* < 0.01) and less likely to perceive higher levels of academic progress (AOR = 0.47, 95% CI = [0.29, 0.77], *p* < 0.01) than students who did not have food insecurity experience in college.

Among the student demographic characteristics, identifying as LGBTQIA++ was associated with experiencing more food stress (AOR = 1.99, 95% CI = [1.08, 3.66], *p* < 0.05) and general stress (AOR = 2.44, 95% CI = [1.34, 4.46], *p* < 0.01), greater difficulty in concentrating on studies (AOR = 2.45, 95% CI = [0.87, 6.92], marginally significant with *p* < 0.1), experiencing more food challenges after the COVID-19 outbreak (AOR = 2.17, 95% CI = [1.03, 4.55], *p* < 0.05), and lower self-rated health (AOR = 0.29, 95% CI = [0.16, 0.53], *p* < 0.01). Being female was associated with more general stress (AOR = 1.60, 95% CI = [1.02, 2.53], *p* < 0.05) and poorer self-rated health (AOR = 0.59, 95% CI = [0.38, 0.93], *p* < 0.05). Being non-Hispanic Black was associated with lower perceptions of academic progress (AOR = 0.31, 95% CI = [0.10, 0.97], *p* < 0.05). These associations were not attenuated after college food insecurity was added (with the exception for race/ethnicity, which was significance decreased from *p* < 0.05 to *p* < 0.1), meaning that childhood food insecurity remained a significant influence on these outcomes for LGBTQIA students and female students.

The *p*-values from the original models and *p*-values corrected for multiple hypothesis testing using Romano–Wolf procedure are reported in Appendix A. With a multiple hypothesis testing correction, the association between childhood food insecurity and three outcomes remained significant at the *p* < 0.05 level in model 1 (without college food insecurity), including food stress, difficulty with concentrating on studies, and self-rated health, and marginally significant for general stress (*p* < 0.1). It was associated with one outcome, difficulty with concentrating on studies, with marginal significance (*p* < 0.1) in model 2 (with college food insecurity added). College food insecurity was still significantly associated with all six outcomes (*p* < 0.01). The associations between other sociodemographic variables and these outcomes became mostly non-significant if the original unadjusted *p*-values were greater than 0.01. The associations that remained at least marginally significant included the association between female and self-rated health in model 2 (*p* < 0.1), the associations between LGBTQIA++ and food stress (model 2, *p* < 0.1), general stress (both models, *p* < 0.05), and self-rated health (both models, *p* < 0.01), and the association between living off campus and food stress (*p* < 0.01 in model 1 and *p* < 0.1 in model 2).

## 4. Discussion and Implications

Our study contributes to the literature on college food insecurity through its focus on the social determinants of health, including sociodemographic characteristics and experiences with childhood food insecurity, the inequities embroiled in those experiences, and the amplified negative impacts on well-being. Overall, the results reveal a more nuanced picture of food insecurity amongst college students, specifically that food insecurity is not isolated from other experiences. Students can be affected by their early life experiences, such as childhood food insecurity, and by the events they are currently experiencing, such as college food insecurity.

Nearly one-fourth of the students in our study identified as food insecure. Consistent with our first hypothesis stating that social demographic characteristics affect college food insecurity, we found that Black students, students living off campus, and first-generation students were more likely to experience food insecurity, and so were in-state students and students majoring in humanities/behavioral/social/health studies. Also consistent with our first hypothesis, our study showed that students who experienced childhood food insecurity were three times more likely to be food insecure in college. The findings also support our second hypothesis stating that food insecurity is negatively associated with college students’ health and well-being and academic progress. Food insecurity was linked with difficulty in concentrating on studies and negatively associated with perceived academic progress, confirming prior research that connected food insecurity with poorer academic performance [55,57]. Regarding stress, food-insecure students faced higher levels of self-rated stress than food-secure students. Our study did not delineate the specific causes of stress for these students, but it was likely a combination of factors related to food insecurity, general academic pressures, and the pandemic, especially as young people have borne the burden of these compounding pressures [76].

Our study results support the idea that as institutions seek to address student needs, students’ prior experiences with food insecurity must be considered. The binary logistic regressions show that students who may already be disadvantaged when they arrive at college (non-Hispanic Blacks and first-generation students, in particular) continue to face disadvantages in college.

Our third hypothesis stating that childhood food insecurity affects college students’ health and well-being and that this effect is mediated by college food insecurity also received support. Childhood food insecurity was associated with five well-being measures (i.e., food stress, difficulty concentrating on studies, more food challenges after the COVID-19 outbreak, general stress, and perceived academic progress) and its association with two outcomes, difficulty concentrating on studies due to food and self-rated health, remained significant after its effect on college food insecurity was accounted for. These patterns generally held with a multiple hypothesis testing correction. Although previous research showed that childhood food insecurity is associated with college food insecurity and college food insecurity is associated with college students’ health and well-being, our study made a unique contribution through mediation analysis by demonstrating that a large proportion of the effect of childhood food insecurity and college students’ health and well-being was explained by college food insecurity, and thus, deepening our understanding of the relationship between childhood food insecurity, college food insecurity, and various health outcomes. These findings further support the notion that disadvantaged statuses accumulate and early life experience can have long-lasting impact on individuals’ later life experience and their health and well-being. These results confirm Camelo and Elliott’s [29] finding that college students are influenced by pre-existing inequities and “histories of unequal opportunities to learn…and uneven access to adequate nutrition” (p. 314).

It should be noted that although the association between childhood food insecurity and perceived academic progress and more food challenges after the COVID-19 outbreak were in the expected direction, childhood food insecurity was not significantly associated with perceived academic progress and not significantly associated with more food challenges after the COVID-19 outbreak after multiple hypothesis testing corrections. These findings may suggest that despite its persistent and enduring effects, childhood disadvantages are not necessarily insurmountable. Great attention to the needs of these students may help them overcome food challenges in college and succeed academically.

### 4.1. Implications of SDOH for University Programming

The SDOH framework is a recommended response to the challenges that contribute to student food insecurity, including experiencing childhood food insecurity, and to the largely individualistic solutions typically recommended for addressing student food insecurity. As universities seek to address food insecurity amongst college students, it is essential to address the distal, structural causes of food insecurity and to understand how the impacts of poverty and food insecurity vary and accumulate across time. Ultimately, inequality itself intensifies these trends, as individuals face compounding challenges in the face of intransigent structural inequalities. More specifically, nutritional disadvantages (and advantages), which are influenced by inequality, can follow an individual throughout their lives [77], not just before and throughout college, but after college attendance. Research shows, for example, that college student food insecurity is associated with the increased prevalence of adult food insecurity after college during early to middle adulthood [78]. Furthermore, attending college does not always improve disadvantaged students’ financial situations [43]. College itself and the subsequent debt for many students can continue this cycle of debt for college students who experience food insecurity, especially as college student debt has lifelong implications, especially for already marginalized students [79].

There is a need to develop policies and programs that more effectively address the holistic dimensions of college student health and its relationship to food insecurity and other challenges. Solutions often focus on the college students’ deficits or only consider individual-level interventions or downstream programs (e.g., food pantries, financial management courses, or cooking classes). Addressing food insecurity on a case-by-case basis and focusing on students’ shortcomings, such as their inability to manage their finances [80], induces shame and stigma and does not tackle the roots of college food insecurity. Many universities provide services to ameliorate food-related challenges, including by requiring first-year students to purchase a meal plan or by providing an on-campus food pantry. However, being on a meal plan does not always result in positive outcomes for students [81]. And students may not willingly use a food pantry due to the stigma associated with the resource, although recent research shows that if structured properly, a food pantry can help to increase a sense of community among students facing food insecurity and help to mitigate its negative impacts on physical and mental health [82]. Increasingly, universities are identifying other forms of support for students, including single-office units and staffing specifically devoted to addressing food and housing insecurity for students with institution-specific programming that helps to alleviate campus-specific needs, as well as partner with national movements, like the Hunger-Free Campus Bill and College Supplemental Nutrition Assistance Program for students not enrolled full time or who meet SNAP exemptions [83].

It is essential for university staff and faculty to understand the connections between SDOH, student identity, college struggles, and childhood experiences [84,85]. Such an approach is aligned with the recent call for universities to give greater attention to creating a culture of health equity and food security on college campuses [86]. Given the compounding and overlapping impacts of food insecurity and far-reaching events, such as the pandemic, efforts to address food insecurity would be best framed within the context of a trauma-informed approach [87,88].

Our findings on the relationship between childhood food insecurity experience and the college food insecurity experiences reinforce prior research [89,90,91,92], indicating a need to consider intersectionality within a life course perspective of food insecurity [33]. This may be challenging, as only one-third of students who experienced struggles with food insecurity, depression, and other challenges reported they shared such experiences with faculty or staff [89].

When assessing students’ pre-college experiences and health-related challenges and assets, institutions should use non-stigmatizing approaches to screen students for prior experiences with food insecurity [92]. Prior research demonstrated the value of utilizing SDOH-specific screening for college students [23]. Screening for SDOH has been implemented in the medical setting [93,94]. One recommendation might be for university on-campus health facilities to include questions related to SDOH on their current prescreening tools. The LIFESCREEN-C screening tool provides a starting point for incorporating SDOH into screening protocols [13], as does the Hunger Vital Sign screening tool [95]. Such examples can serve as a basis for developing a screening tool specifically tailored for college students that provides a nuanced understanding of food insecurity and other challenges. Universities may also require a SDOH prescreen at first-year mandatory programming events, such as orientations, courses, or online training sessions. Implementing SDOH measures on intake forms for students can be easy to implement [58,96] and can help administrators identify and assess SDOH in the student population, thus providing direction on implementing programming that may mitigate barriers to student success. 

Research suggests that students have limited understanding of issues pertaining to health equity and SDOH [97]. Thus, when considering implications for health education for college students, it is also important for university staff to educate students about the specific SDOH that impacts their college experiences. This knowledge can empower students to apply this knowledge to their own lives, including understanding how prior experiences with various SDOH may impact current and future health outcomes. At a broader level, interventions that incorporate SDOH and structural competence include both improving access to healthy and affordable food (SDOH) [73] and advocating for broadening SNAP benefits for college students and supporting an increase in Pell grants (structural competency), as was recently announced for President Biden’s 2023 fiscal plan [98].

### 4.2. Limitations

While the study results yield unique insights into food insecurity among college students, our study had several limitations. Our study relied on convenience sampling at one institution, precluding the ability to generalize to other settings. However, our findings are consistent with existing literature and extend the conversation in new directions with the focus on childhood food insecurity. As we used a cross-sectional design, the direction of relationships between the variables cannot be established. For our statistical analysis, where statistically significant results are reported (e.g., for Black students), the small cell sizes for some of the demographic variables may reflect spurious or overfitted findings. Our use of the Hunger Vital Sign for college students and in a retrospective fashion was used in prior studies on college food insecurity [70,71]; future research could continue to include such measures and approaches to more fully investigate the impact of childhood food insecurity on young adult and adult food insecurity. However, student responses to the various survey items may have been influenced by potential recall bias, as they were asked to reflect on food insecurity experienced during their childhood. Our measures of health and well-being were derived from our literature review, discussed in an earlier section of the paper, on college students’ experiences overall. The items were drawn from other instruments, and thus, further exploration is warranted to better understand the nuances of the impact of food stress and food insecurity on college student experiences.

Our quantitative approach could be enhanced with longitudinal research and qualitative studies that can highlight the nuances of college student food insecurity, including differences by key sociodemographic characteristics. Research showed, for example, that it is essential to consider the intersectional nature of cumulative disadvantages over one’s life course [33], the cumulative burden of experiencing other needs during college [99], and the intersectional nature of academic achievement and food insecurity, beginning with K-12 education [29]. We recommend qualitative research about students’ lived experiences with food insecurity and other challenges, which is a point supported by Ezarik [89], who encouraged “connecting with students by hearing their stories”. We recommend that future studies include measures regarding whether students have access to SNAP or Pell grants or other scholarships, as these structural programs can potentially mitigate food insecurity. Future studies should also include a measure regarding whether students received SNAP benefits during childhood, as prior research has shown that the receipt of SNAP benefits during childhood reduces the impact of childhood food insecurity on health outcomes experienced as an adult [38]. Finally, we recommend that future studies more fully consider whether the USDA measure of food insecurity adequately captures the complexity of college students’ food experiences, which was a concern raised in prior research [100,101,102]. As the USDA measure pertains to household living situations of individuals who are related to one another, it may not adequately capture the living arrangements and the health and well-being of college students.

## 5. Conclusions

Our study showed that food insecurity among college students was about more than food; it was influenced by sociodemographic characteristics and had temporal dimensions. College food insecurity was impacted by prior experiences with food insecurity and was related to students’ perceptions of physical and mental health, feelings about academic progress, and difficulty in concentrating on their studies. College food insecurity also had spatial dimensions in that the lived experiences of college students were shaped by the institutional context of higher education in which students were struggling with diverse challenges, including higher tuition costs and the subsequent debt for many.

The SDOH framework, with its focus on structural competency, encapsulates the structural causes of food insecurity of college students, including cumulative disadvantages resulting from childhood food insecurity. These structural concerns are broad and larger than the institutions themselves. Addressing the broader causes and context of student vulnerabilities in general, and food insecurity in particular, helps to destigmatize poverty and food insecurity and increases student engagement [41]. In this way, rather than reproducing inequality, higher education can ultimately be the engine of social opportunity that it is designed, or at least purports itself, to be.

## Figures and Tables

**Table 1 nutrients-16-01391-t001:** Descriptive statistics for all and by college food insecurity status.

Variables	All (*N* = 372)	Food Secure (*N* = 284)	Food Insecure (*N* = 88)	χ^2^ (*p*) for Difference by Food Insecurity
College food insecurity	23.7% (88)			
Female	73.1% (272)	75.4% (214)	65.9% (58)	3.048 (*p* = 0.081)
LGBTQIA++	11.6% (43)	10.2% (29)	15.9% (14)	2.134 (*p* = 0.144)
Race/ethnicity				9.201 (*p* = 0.056)
Non-Hispanic White	76.6% (285)	78.9% (224)	69.3% (61)	
Non-Hispanic Black	3.0% (11)	1.8% (5)	6.8% (6)	
Other race/ethnicity, non-Hispanic	4.3% (16)	3.5% (10)	6.8% (6)	
Multiracial	8.1% (30)	8.5% (24)	6.8% (6)	
Hispanic	8.1% (30)	7.4% (21)	10.2% (9)	
Living off campus	55.4% (206)	50.7% (144)	70.5% (62)	10.605 (*p* = 0.001)
In-state student	61.0% (227)	56.3% (160)	76.1% (67)	11.072 (*p* = 0.001)
First-generation college student	10.8% (40)	7.8% (22)	20.5% (18)	11.306 (*p* = 0.001)
Study area				8.519 (*p* = 0.014)
Agriculture/engineering/science	33.3% (124)	35.6% (101)	26.1% (23)	
Humanities/behavioral/social/health/education	46.8% (174)	42.6% (121)	60.2% (53)	
Business	19.9% (74)	21.8% (62)	13.6% (12)	
Childhood food insecurity	12.1% (45)	7.4% (21)	27.3% (24)	24.966 (*p* < 0.001)
Food stress (1–4), *mean (std)*	1.86 (0.96)	1.57 (0.80)	2.78 (0.84)	108.345 (*p* < 0.001)
None	47.3% (176)	59.9% (170)	6.8% (6)	
Less than average	25.5% (95)	25.0% (71)	27.3% (24)	
Average	21.0% (78)	13.0% (37)	46.6% (41)	
More than average	6.2% (23)	2.1% (6)	19.3% (17)	
Difficulty concentrating on studies	6.5% (24)	1.8% (5)	21.6% (19)	43.774 (*p* < 0.001)
More food challenges after COVID-19 outbreak	18.3% (68)	14.1% (40)	31.8% (28)	14.144 (*p* < 0.001)
General stress (1–4), *mean (std)*	2.68 (0.79)	2.58 (0.76)	3.00 (0.82)	26.314 (*p* < 0.001)
Less than average	6.7% (25)	7.8% (22)	3.4% (3)	
Average	32.0% (119)	34.9% (99)	22.7% (20)	
More than average	47.6% (177)	48.6% (138)	44.3% (39)	
Extreme	13.7% (51)	8.8% (25)	29.6% (26)	
Self-rated health (1–4), *mean (std)*	2.50 (0.87)	2.60 (0.84)	2.17 (0.89)	18.882 (*p* < 0.001)
Poor/fair	12.9% (48)	9.5% (27)	23.9% (21)	
Good	36.8% (147)	34.9% (99)	43.2% (38)	
Very good	37.6% (140)	41.6% (118)	25.0% (22)	
Excellent	12.6% (47)	14.1% (40)	8.0% (7)	
Perceived academic progress (1–4), *mean (std)*	3.43 (0.76)	3.49 (0.74)	3.23 (0.81)	9.831 (*p* < 0.020)
Poor	2.7% (10)	2.5% (7)	3.4% (3)	
Fair	8.6% (32)	7.0% (20)	13.6% (12)	
Good	31.7% (118)	29.2% (83)	39.8% (35)	
Excellent	57.0% (212)	61.3% (174)	43.2% (38)	

Note: Ns are in parentheses.

**Table 2 nutrients-16-01391-t002:** Adjusted odds ratios (AORs) from regressions of college food insecurity (*N* = 372).

Variables	Model 1	Model 2
Female	0.686	0.675
	[0.390, 1.207]	[0.379, 1.200]
LGBTQIA++	1.132	1.007
	[0.539, 2.378]	[0.468, 2.166]
Race/ethnicity (ref = non-Hispanic White)		
Non-Hispanic Black	3.438	2.642
	[0.898, 13.161]	[0.670, 10.425]
Other race/ethnicity, non-Hispanic	2.180	2.100
	[0.731, 6.505]	[0.679, 6.497]
Multiracial	0.783	0.891
	[0.287, 2.133]	[0.326, 2.440]
Hispanic	1.532	1.525
	[0.604, 3.882]	[0.594, 3.919]
Living off campus	1.980 *	1.962 *
	[1.140, 3.442]	[1.119, 3.441]
In-state student	1.820 *	1.605
	[1.016, 3.260]	[0.886, 2.908]
First-generation college student	2.285 *	1.774
	[1.076, 4.849]	[0.796, 3.954]
Study area (ref = agriculture/engineering/science)		
Humanities/behavioral/social/health	2.090 *	2.093 *
	[1.156, 3.777]	[1.147, 3.820]
Business	1.107	1.132
	[0.493, 2.488]	[0.500, 2.560]
Childhood food insecurity		3.166 **
		[1.541, 6.507]
Chi-square	40.19	49.81
*df*	11	12

Notes: binary logistic regressions were estimated; 95% confidence intervals are in parentheses; ** *p* < 0.01, * *p* < 0.05 (two-tailed tests).

**Table 3 nutrients-16-01391-t003:** Adjusted odds ratios (AORs) from regressions of food stress, difficulty concentrating on studies, and more food challenges after COVID-19 outbreak on student characteristics, childhood food insecurity, and college food insecurity (*N* = 372).

Variables	Food Stress		Difficulty Concentrating on Studies		More Food Challenges after COVID-19 Outbreak	
	Model 1	Model 2	Model 1	Model 2	Model 1	Model 2
Female	1.033	1.351	0.853	1.093	1.191	1.281
	[0.660, 1.617]	[0.845, 2.159]	[0.325, 2.235]	[0.378, 3.156]	[0.624, 2.272]	[0.662, 2.478]
LGBTQIA++	1.986 *	2.184 *	2.452	3.258 *	2.165 *	2.217 *
	[1.077, 3.662]	[1.172, 4.072]	[0.869, 6.917]	[1.020, 10.410]	[1.029, 4.553]	[1.042, 4.717]
Race/ethnicity (ref = non-Hispanic White)						
Non-Hispanic Black	1.454	1.025	0.549	0.235	1.686	1.389
	[0.495, 4.265]	[0.337, 3.124]	[0.056, 5.382]	[0.020, 2.777]	[0.433, 6.560]	[0.335, 5.768]
Other race/ethn, non-Hisp	1.785	1.454	1.617	0.895	0.579	0.466
	[0.726, 4.388]	[0.557, 3.796]	[0.292, 8.962]	[0.126, 6.363]	[0.124, 2.696]	[0.096, 2.274]
Multiracial	0.925	0.967	0.593	0.570	0.317	0.321
	[0.437, 1.955]	[0.440, 2.126]	[0.067, 5.271]	[0.057, 5.710]	[0.070, 1.424]	[0.071, 1.453]
Hispanic	1.349	1.087	1.616	1.725	1.736	1.649
	[0.628, 2.899]	[0.495, 2.387]	[0.376, 6.944]	[0.383, 7.774]	[0.703, 4.288]	[0.662, 4.105]
Living off campus	2.004 **	1.727 *	2.521	1.845	1.250	1.158
	[1.324, 3.033]	[1.126, 2.649]	[0.840, 7.561]	[0.582, 5.848]	[0.697, 2.240]	[0.639, 2.100]
In-state student	1.301	1.149	1.426	1.200	0.786	0.722
	[0.845, 2.002]	[0.738, 1.791]	[0.472, 4.310]	[0.371, 3.886]	[0.433, 1.428]	[0.392, 1.329]
First-generation college student	1.174	0.943	2.143	1.791	1.598	1.458
	[0.604, 2.283]	[0.468, 1.900]	[0.690, 6.657]	[0.541, 5.925]	[0.686, 3.723]	[0.618, 3.437]
Study area (ref = agric/engin/science)						
Humanities/behavioral/social/health	1.313	0.955	1.131	0.823	0.976	0.871
	[0.840, 2.050]	[0.601, 1.517]	[0.401, 3.191]	[0.275, 2.461]	[0.531, 1.792]	[0.467, 1.623]
Business	1.191	1.070	1.319	1.181	0.741	0.742
	[0.679, 2.088]	[0.598, 1.913]	[0.336, 5.170]	[0.252, 5.550]	[0.324, 1.698]	[0.321, 1.716]
Childhood food insecurity	2.618 **	1.635	5.245 **	3.884 **	2.002	1.637
	[1.392, 4.923]	[0.864, 3.092]	[1.973, 13.948]	[1.387, 10.873]	[0.932, 4.303]	[0.747, 3.590]
College food insecurity		10.866 **		11.550 **		2.540 **
		[6.367, 18.544]		[3.806, 35.048]		[1.361, 4.739]
Chi-square	45.91	130.9	29.96	52.72	21.20	29.53
*df*	12	13	12	13	12	13

Notes: ordered logistic regressions were estimated; 95% confidence intervals in brackets; ** *p* < 0.01, * *p* < 0.05 (two-tailed test).

**Table 4 nutrients-16-01391-t004:** Adjusted odds ratios (AORs) from regressions of general stress, self-rated health, and perceived academic progress on student characteristics, childhood food insecurity, and college food insecurity (*N* = 372).

Variables	General Stress		Self-Rated Health		Perceived Academic Progress	
	Model 1	Model 2	Model 1	Model 2	Model 1	Model 2
Female	1.602 *	1.723 *	0.590 *	0.543 **	1.202	1.135
	[1.016, 2.525]	[1.089, 2.725]	[0.375, 0.930]	[0.343, 0.862]	[0.756, 1.912]	[0.711, 1.811]
LGBTQIA++	2.444 **	2.544 **	0.286 **	0.275 **	0.776	0.788
	[1.339, 4.460]	[1.390, 4.657]	[0.155, 0.527]	[0.150, 0.507]	[0.405, 1.485]	[0.410, 1.517]
Race/ethnicity (ref = non-Hisp White)						
Non-Hispanic Black	1.445	1.183	0.785	0.918	0.311 *	0.352
	[0.483, 4.320]	[0.387, 3.613]	[0.243, 2.541]	[0.275, 3.062]	[0.099, 0.972]	[0.110, 1.124]
Other race/ethn, non-Hisp	1.166	0.990	1.135	1.174	0.744	0.849
	[0.443, 3.066]	[0.381, 2.571]	[0.449, 2.871]	[0.464, 2.971]	[0.285, 1.943]	[0.323, 2.228]
Multiracial	1.155	1.167	1.022	1.015	0.871	0.852
	[0.573, 2.331]	[0.576, 2.366]	[0.501, 2.086]	[0.499, 2.063]	[0.409, 1.853]	[0.400, 1.815]
Hispanic	1.739	1.649	0.850	0.903	0.511	0.552
	[0.819, 3.691]	[0.766, 3.553]	[0.417, 1.732]	[0.441, 1.848]	[0.240, 1.089]	[0.259, 1.177]
Living off campus	1.196	1.064	0.846	0.911	0.812	0.884
	[0.797, 1.794]	[0.705, 1.604]	[0.568, 1.259]	[0.610, 1.360]	[0.528, 1.249]	[0.571, 1.369]
In-state student	0.973	0.930	0.934	0.982	1.028	1.063
	[0.639, 1.482]	[0.610, 1.419]	[0.617, 1.414]	[0.647, 1.490]	[0.657, 1.609]	[0.678, 1.668]
First-generation college student	0.881	0.775	1.275	1.398	0.854	0.924
	[0.453, 1.715]	[0.395, 1.518]	[0.662, 2.459]	[0.722, 2.707]	[0.429, 1.701]	[0.461, 1.849]
Study area (ref = agric/engin/science)						
Humanities/behavioral/social/health	0.915	0.805	1.045	1.125	1.569	1.770 *
	[0.588, 1.426]	[0.514, 1.262]	[0.677, 1.613]	[0.726, 1.744]	[0.983, 2.506]	[1.096, 2.859]
Business	0.902	0.868	1.388	1.401	0.595	0.597
	[0.522, 1.558]	[0.501, 1.503]	[0.808, 2.384]	[0.814, 2.412]	[0.338, 1.045]	[0.340, 1.048]
Childhood food insecurity	2.025 *	1.617	0.386 **	0.491 *	0.759	0.864
	[1.097, 3.739]	[0.864, 3.028]	[0.211, 0.704]	[0.264, 0.913]	[0.390, 1.479]	[0.442, 1.686]
College food insecurity		2.801 **		0.420 **		0.468 **
		[1.666, 4.707]		[0.257, 0.686]		[0.285, 0.770]
Chi-square	23.39	38.89	38.38	50.54	23.05	31.97
*df*	12	13	12	13	12	13

Note: ordered logistic regressions were estimated; 95% confidence intervals in brackets; ** *p* < 0.01, * *p* < 0.05 (two-tailed test).

## Data Availability

Due to privacy and ethical concerns, neither the data nor the source of the data can be made available.

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
