# Peer review of "Social Determinants of Health and College Food Insecurity"

_nutrients, 2024, doi:10.3390/nu16091391_

Round 1

Reviewer 1 Report

Comments and Suggestions for Authors

This is a single site study of self-reported food insecurity among 372, predominantly Non Hispanic White female students at single Southeastern university.  Data were collected during the waning of The COVID epidemic in fall 2021. The sample was essentially volunteer students with snowball recruitment  who responded to a Qualtrics survey   without any particular sampling restrictions or targets.  The paper, which is very well organized and clearly written, demonstrates clear awareness of current theory and literature. The article addresses an interesting and timely issue, but has non-trivial limitations which the authors need to delineate better.

Typo line102  “college good insecurity for “college food insecurity.

Line 111 Should be self identified male or female

Lines 172  the concept of nutrition insecurity should be defined in the first paragraph if the authors feel it is relevant to stressors experienced by students above and beyond food insecurity – if it is not, the construct (which is not directly measured by study instruments and for which measurement tools are still evolving) could  be omitted. The authors are referred to https://www.healthaffairs.org/do/10.1377/hpb20230216.926558/

Line245-254  The retrospective framing of the widely used Hunger Vital Sign has not, as far as this reviewer, knows been validated or used in other research. (The citations are for current use in varying populations.) When it appears throughout the text it should be termed “exploratory” or similar term to make clear that this is a new usage. Retrospective assessment of food insecurity is  an intriguing idea, but the discussion should make clear that this is a non-standard, non-validated use of the instrument whose test retest reliability as a retrospective measure had not been described.

Lines 255-82 the queries all seem to have face validity , but seem to be study specific. If they are drawn from other instruments or their psychometric properties evaluated in other studies this should be so stated. If not the discussion should address this as a limitation. The concept of food stress seems to appear as a statistically significant predictor when the better validated six item food insecurity scale does not, which warrants further explication.

Line 305 “significant” should state statistically significant (as the authors appropriately do in the subscript to Table 1) and specify whether the calculations were one or two tailed.

357 don’t need percents except for press conferences!  Scientific report should be should be adjusted odd ratio AOR and 95% CI > Tabe 2 should specify adjusted odds ratio. Confidence intervals are wide and scrutiny of Table I indicates that many cell size were small. It would be helpful to give the n of each category after the label e.g female  (271) Black (11) etc to assist in in the interpretation of Table 2.

Lines 429 and throughout – be careful about reporting race/ethnicity Black should always be capitalized and the social construct of race should always be term “race/ethnicity”

Lines 433 and following it is helpful that the authors present correction for multiple hypotheses

4.1 Authors appropriately allude to the importance of SNAP policies and Pell grants (and presumably other scholarships) for students. They do not discuss or measure in their sample whether students had access to these programs or to term time jobs which are structural measures which might influence risk of food insecurity.

4.2 appropriately discusses some limitations but omits others:  unvalidated exploratory measures (including the retrospective Hunger Vital Sign,)  the queries in lines 252 following, potential recall bias of childhood food insecurity, small cell size in demographic domains where statistically significant results are reported (e.g. Black), which may be spurious or overfitted.

The authors may also wish to allude to the implication of this work for future scientific efforts. In spite of its limitations, this paper suggests that before drawing conclusions about the irreversible impact of childhood experiences years later, it is important to identify concurrent (and thus potentially reversible) conditions which may better explain current findings.

Reviewer 2 Report

Comments and Suggestions for Authors

Relevant and useful study as college food insecurity is a growing issue in the U.S.

Minor changes required.

Line 10 and 35: Oddly phrased sentence.

Line 38: Suggest citing nationally representative rates. “The first national-level, representative survey of college students shows that 22.6% of undergraduates experience food insecurity, with another 11.9% experiencing marginal food security (National Center for Education Statistics, 2023). Pg 3, DOI: 10.3102/00028312231217751.  

Line 47: cite the SDOH framework

Line 51: CoFI abbreviation is used in the abstract, but not in this paragraph

Line 63-65: “much research’ is not cited.

Line 480: add the 5 measures to the sentence to make it clear.

Comments on the Quality of English Language

Few sentences need to be checked.

Line 10 and 35: Oddly phrased sentence.

Round 2

Reviewer 1 Report

Comments and Suggestions for Authors

The authors have made a good faith effort to respond to the previous review, but some issues remain.

Abstract needs to specify that this is a convenience sample and the dates of data collection (Fall 2021)

Line 35 More than what?

Line 40-43 Spring 2020 was during COVID -were these data covering the previous year or another interval such as those used in the papers summarized in lines 182-91?  Line 246 specifies last 30 days which should be noted in the abstract

Line 205-206 (hypothesis H3) should specify retrospectively ascertained childhood food insecurity. Since previously reviewing this paper the current reviewer has found a study based on the PSID which provides truly prospective longitudinal  assessment of adult outcomes of food insecurity in childhood using the full 18 item USDA Household food security scale (Fertig, A. (2019, May). The long-term health consequences of childhood food insecurity. University of Kentucky Center for Poverty Research Discussion Paper Series, DP2019-03. Retrieved [Date] from http://ukcpr.org/research.) and showing increased psychological distress measured in young adulthood following experiencing food insecurity in childhood controlling for adult food insecurity.  The authors should consider this paper which provides useful context supporting their exploratory findings using the Hunger Vital Sign retrospectively.

Line 421 and ff were these adjusted odds ratios (in which case they should be termed AOR) or unadjusted. If adjusted the covariates used in the adjustment should be specified.

Line 603 there is a random letter n the sentence

Line 606 and following The authors write “Our measures of health and well-being were derived from our literature review, discussed in an earlier section of the paper, on college students’ experiences overall. The items are drawn from other instruments and thus further exploration is warranted to better understand the nuances of the impact of food stress and food insecurity on college student experiences.”

This Is still quite vague – ideally each measure should have a bibliographic reference  or references cited when the measure is first introduced in sections 2.2.3 through 2.2.8.
